# Is Beauty a Matter of Volume Distribution? Proposal of a New Aesthetic Three-Dimensional Guide in Orthognathic Surgery

**DOI:** 10.3390/jpm13060936

**Published:** 2023-06-01

**Authors:** Alberto Bianchi, Francesco Seidita, Giovanni Badiali, Laura Lusetti, Cristiana Saporosi, Marco Pironi, Claudio Marchetti, Salvatore Crimi

**Affiliations:** 1Maxillofacial Surgery Unit, “Policlinico-San Marco” University Hospital, Viale Azeglio Ciampi, 95121 Catania, Italy; alberto.bianchi@unict.it (A.B.); torecrimi@gmail.com (S.C.); 2Maxillofacial Surgery Unit, “Sant’Orsola” University Hospital, Via Giuseppe Massarenti 9, 40138 Bologna, Italy; giovanni.badiali@unibo.it (G.B.); claudio.marchetti@unibo.it (C.M.); 3Plastic Surgery Unit, “Policlinico di Modena” Hospital, Via del Pozzo 71, 41100 Modena, Italy; 4Independent Researcher, 40100 Bologna, Italy; cristiana.saporosi@gmail.com; 5Independent Researcher, 40100 Bologna, Italy; mp@marcopironi.it

**Keywords:** orthognathic surgery, cephalometry, orthognathic surgical procedure, maxillo-mandibular surgery, aesthetics dental, diffusion innovation, transfers technology

## Abstract

Background: Orthognathic surgery is a multidisciplinary surgery in which the aesthetic results have become increasingly important, and consequently, also the predictability of the surgical outcomes. In this paper, we analyzed the volumetric distribution of the lower two-thirds of the face, in patients operated by orthognathic surgery and selected for their attractiveness. Our goal was to analyze the aesthetic volumetric distribution for gender and to propose our operating philosophy, that a normative distribution of facial volumes could be used like a new 3D aesthetic guide in orthognathic planning. Methods: A group of 46 orthognathic patients (26 females, 20 males) with the best postoperative aesthetic score was selected by a jury of plastic surgeons, orthodontists, and journalists. The mean soft tissue volumes of the malar, maxillary, mandibular, and chin regions were analyzed. Results: Overall, we measured a mean female facial volume distribution of 38.7%, 29%, 27.6%, and 4.7%, respectively, in the malar, maxillary, mandibular, and chin regions, while in males, it was 37%, 26%, 30%, and 6%, respectively. Conclusions: In this paper, the expansion of facial volumes in orthognathic surgery is proposed as a key point for facial harmonization. Beauty could be scientifically interpreted as a balanced distribution of facial volumes, and the virtual study of this distribution can become an important part of the preoperative analysis, such as a “volumetric” 3D cephalometry, where the surgeon could use average values of aesthetic volumetric distribution as preoperative surgical references.

## 1. Introduction

Orthognathic surgery has gone through a progressive evolution over time. Born as an occlusal-only surgery, it has now become a multidisciplinary surgery. Modern orthognathic surgery aims at several goals: occlusal restoration, respiratory improvement, phonatory, etc. [1]. One of the main goals of modern orthognathic surgery is facial aesthetic harmonization [2], which can be considered one of the most driving matters for patients.

Although the concept of beauty may seem abstract or subjective, there are some aesthetic canons in which beauty resides. This is true for nature, anthropometry, and consequently, for cephalometric and facial analysis [3].

With the surgical philosophy of facial skeletal expansion, in 1992 H. M. Rosen [4] introduced the concept that aesthetic improvement is proportional to the transverse and sagittal facial expansion achieved. An increase in maxillary, mandibular, and chin protrusion (named “excessive anterior divergence”), with an adequate transverse expansion, is the main aesthetic goal [4]. However, this milestone in modern orthognathic surgery remains anchored to bidimensional considerations such as sagittal expansion and transverse expansion. We promote the evolution of this idea, proposing it in a new three-dimensional view, in terms of volumetric redistribution.

In this paper, we analyze how orthognathic surgery has evolved, especially concerning the transition to the third dimension and the improving accuracy of Virtual Surgical Planning (VPS). At the same time, we show how the aesthetic demands of the patients are increasing with special regard to soft tissue aesthetic outcomes.

This new background requires a new way of planning cases, focusing on the aesthetic soft tissue volumetric redistribution, since the software is now accurate enough to predict the volumetric outcome. In fact, although the goal of soft tissue volumetric redistribution is widely accepted in aesthetic surgeries in general and also in modern orthognathic surgery, to the best of our knowledge, there are no real practical applications at the moment of the VSP; in particular, the current three-dimensional cephalometry does not provide for easily analyzing the areas of volumes to be corrected. Moreover, it is not yet disclosed how the expected volumes can influence the choice of bone movements.

To plan a case based on how volumes can be redistributed, what we need is a standard reference of “ideal” or “aesthetic” volumetric distribution for gender and ethnicity. For this purpose, we analyzed the mean volumes of 4 facial regions (malar, maxillary, mandibular, and chin) in a group of 46 orthognathic patients with the best postoperative aesthetic score, selected for their final attractiveness.

In this paper, we show our aesthetic selection method and our method for volumetric analysis. The aim of our study is not only to show our volumetric references, which we recommend for the reasons we discuss since each author can choose his own. In fact, we also want to promote and discuss our procedural philosophy that a balanced volume redistribution is the key to facial harmonization, and to introduce the volumetric cephalometry as a possible new aesthetic guide for VSP. We believe that the software for VSP will expand their tools, making the volumetric analyses easier.

## 2. Patients and Methods

This paper is a retrospective observational trial, conducted in accordance with the STROBE Statement checklist of items guidelines [5].

This is a bi-centric study, performed between March and September 2021 at the Maxillofacial Surgery Units of San Marco University Hospital (Catania, Italy) and Sant’Orsola University Hospital (Bologna, Italy). During this period, we retrospectively studied all patients previously operated on by orthognathic surgery in the two Centers starting in 2016.

The inclusion criteria for the trial were: adult patients of the Caucasian race, undergoing orthognathic surgery (with or without genioplasty), regardless of the type of malocclusion, with a postoperative follow-up of at least 6 months, including postoperative Cone Bean Computed Tomography (CBCT) and complete photographic study. Minor patients, patients with malformations, and patients undergoing postoperative aesthetic ancillary procedures were excluded from the trial.

This study was performed according to the Helsinki Declaration and in agreement with the Local Ethics Committees of the two hospitals (authorization 351/2017/O/OssN), which only required patients to sign an informed consent to authorize the retrospective study of their data, with anonymously collected data.

Overall, 155 patients who met our selection criteria were enrolled. For each of them, postoperative photographs were subjected to aesthetic judgment by an unconditional jury. As a jury, we have enrolled 6 volunteers: 2 orthodontists, 2 plastic surgeons, and 2 journalists.

For all patients, each judge individually analyzed the postoperative photographs and attributed an aesthetic score. We used the 1–10 aesthetic score according to Phillips, in which 1 corresponded to “not at all attractive” and 10 to “very attractive” [6]. In order not to influence aesthetic judgment, the preoperative photos were not shown.

For all patients, each judge repeated the judgment after 3 weeks, and a Pearson’s correlation test was used to assess the repeatability of each judge’s judgment.

An average aesthetic score greater than or equal to 8.5 was chosen as a cut-off to select the most attractive. Overall, 46 patients (20 males and 26 females) passed the test, becoming the aesthetic subsample of our study.

For the analysis of facial volume distribution, CBCT Digital Imaging and Communications in Medicine (DICOM) files have been converted to Stereolithography interface format (STL) files and then imported into the software MeshMixer (MeshMixer Version 3.3 for Windows, Autodesk Corp., Mill Valley, CA, USA). For each of the 46 heads, both the skeleton and the soft tissues of the face were imported and superimposed.

To obtain the lower two-thirds of each head, the parts of interest in orthognathic surgery, we performed two initial segmentations: as the uppercut, an axial cutting plane passing through the Frankfort Horizontal plane (FH) has been chosen; as the posterior cut, a coronal cutting plane passing through the two Porions (Figure 1).

Each lower two-thirds were further segmented into 4 slices: malar, maxillary, mandibular, and chin.

To segment the malar slice, a cutting plane passing through the bispinal plane (from anterior nasal spine to posterior nasal spine) was chosen (Figure 2a). To segment the maxillary slice, a cutting plane passing through the occlusal plane was chosen (Figure 2b). To segment the mandibular slice and separate it from the neck, a cutting plane passing through the mandibular base (Gonion–Menthon) was chosen (Figure 2c). Finally, to separate the chin from the rest of the mandible, a cutting plane passing through the mental foramina and parallel to FH was used (Figure 2d). Overall, we segmented 4 contiguous facial slices on the CTs, which are represented in Figure 3. Figure 4 shows a representation of the same slides, reported on the 3D photogrammetry of the same patient.

For all 46 patients, using the same software, the volume (mm^3^) of the soft tissues corresponding to each slice was measured and collected in an Excel database (Excel for Windows 2019, Microsoft Corporation, Washington, DC, USA). The mean soft tissue volumes of each facial slide were calculated for gender and compared by proportional ratio.

The statistical analysis was carried out using the *SPSS* software (IBM SPSS Statistics for Windows, Version 20.0. IBM Corp., Armonk, NY, USA). The repeatability of the judgments of each judge was studied by the Pearson correlation test. For both genders, the normality distribution of the volumetric data was studied by Shapiro–Wilk test. For each facial slice, the mean volumes were reported in mm^3^ and also as a percentage of the total volume of the lower two-thirds. Values of *p* ≤ 0.05 were considered statistically significant.

## 3. Results

All data showed a normal distribution (Shapiro–Wilk: 0.950–0.980, *p* > 0.05). Pearson’s correlation test showed no statistically significant differences between the judges’ first and second judgments, and all six judges of the jury were considered reliable (mean *p* value = 0.71, range 0.6–0.8).

In females, the following mean volumetric values were measured: 283,005 mm^3^ for the malar slice (38.7%), 212,433 mm^3^ for the maxillary slice (29%), 202,279 mm^3^ for the mandibular slice (27.6%), and 34,340 mm^3^ for the chin (4.7%). In males, the following mean volumetric values were measured: 300,436 mm^3^ for the malar slice (37%), 230,526 mm^3^ for the maxillary slice (26%), 242,905 mm^3^ for the mandibular slice (30%), and 49,444 mm^3^ for chin (6%).

Our proposal for a new aesthetic three-dimensional guide in orthognathic surgery was presented at the 26th Congress of the European Association of Cranio–Maxillofacial Surgery (September 2022).

## 4. Discussion

In our observational study, we analyzed some facial volumetric proportions in a group of attractive patients who underwent orthognathic surgery, with the idea of promoting the surgical philosophy of reinterpreting facial aesthetics as a distribution of volumes.

Since Margaret Wolfe Hungerford stated that “beauty is in the eye of the beholder” [7], several studies have tried to prove otherwise [8]. Beauty is not just an abstract and subjective concept: in nature, there are some aesthetic canons, which can be mathematically interpreted and can also be applied to human faces. The most discussed is the golden ratio [9,10,11,12,13]. The cephalometric examination itself, understood as a study of facial harmony, is a set of geometries and proportions [14,15].

The first facial aesthetic canons are dated back to the ancient Egyptians and then rearranged during the classical and neoclassical periods [16,17]. Over the centuries, aesthetic canons have changed and have been studied in various forms of art and medicine. To date, the concept of beauty is very complex and the tendency is to also consider the psychological, evolutionary, cultural, and social aspects [18]. After the revolutionary studies of Farkas about the canons for aesthetic facial surgery [19] and after some subsequent studies on the harmony of the maxilla–mandibular complex [20], the current beauty criteria are no longer such simple proportions: they are more complex concepts, such as “symmetry, averageness, sexual dimorphism, or neoteny” [21].

For the above-mentioned reasons, modern orthognathic surgery has been extremely revolutionized: the soft tissue paradigm [22], which highlights the importance of soft tissue adaptation and soft tissue contours, has replaced the angle paradigm [23], in which everything was secondary to an “ideal” occlusion. The aesthetic outcome has become fundamental, not only for the increased demands of patients but rather because the correct functions are found in the correct harmony. Nowadays, orthognathic surgery is considered a “full face” surgery, where aesthetics and function go “hand in hand”: it is a surgery for the soft tissues and not just an occlusal surgery.

Another crucial revolution is the full 3D preoperative planning that is routinely carried out for orthognathic surgery, and with the development of 3D cephalometry, the attention toward soft tissues has further increased [24,25]. The software for Virtual Surgical Planning (VSP) can study every single facial unit in shape, size, position, orientation, and symmetry [26,27], and predictability levels are increasingly accurate for soft tissue as well [28,29].

All the above-mentioned background is fully integrated into our surgical philosophy consisting of reinterpreting beauty as a distribution of volumes.

However, to the best of our knowledge, despite the numerous digressions on the aesthetic canons, there is no correlation in the actual literature between the volume distribution and the aesthetic outcomes after orthognathic surgery. Some papers analyze the correlation between the attractiveness of faces and the protrusion of the chin [30]; others highlight the aesthetics of the profile with a prominence of the lower third [31].

The first author to introduce the concept of “facial expansion” was H. M. Rosen. He believed that the increase in terms of volume of the lower third, both transversely and in profile, could become an aesthetic correction strategy for two main reasons. The first one is that traditional cephalometric standards are not necessarily the most aesthetic. In fact, contrary to traditional cephalometric features, an increase in sagittal cephalometric values beyond the standards can be considered attractive, as a sign of protrusion of the jaws and lip support. The second reason is that the prediction of the results is greater by expanding the defective anatomical regions, rather than reducing the excessive ones, due to the better skeletal support.

According to our three-dimensional reinterpretation of this facial expansion theory, an increase in frontal and/or sagittal dimension determines the increase in a specific volume that can be measured on CBCTs.

From our analysis, the mean malar volume was 37% in males and 38.7% in females, and the mean chin volume was 6% in males and 4.7% in females. It is interesting to note that, also for women, a large volume in the malar and chin region was considered to be attractive. On the other hand, the mean volumes of the maxillary and mandibular regions were reversed in the two sexes, respectively, with 29% and 27.6% for males (maxilla/mandible ratio = 1.05) versus 26% and 30% for females (maxilla/mandible ratio = 0.87). Overall, a volumetric prevalence of the mandibular region over the maxilla was confirmed to be the main feature of masculinization, while the aesthetic projection of the cheekbones and chin was similar in both sexes.

Our study analyzes the facial aesthetic volumetric distributions in both sexes. However, our goal was not to make just a male/female comparison, since it is obvious that faces are different in the two genders. Likewise, it is not our purpose to find some average differences between attractive and not, because it is clear that beauty is individual and not just a mathematical pattern. We have arbitrarily chosen to measure the volume in these four regions because of the strict relation to the bone segments that are normally moved during surgery. Anyhow, potentially, a detailed mapping of the mean volumetric distribution could be done in attractive people. Our purpose was basically to show that this volumetric standard could be used as a reference in the VSP for each ethnicity. In fact, just like a preoperative 2D or 3D cephalometric analysis of a patient is normally compared to the references, in the same way, a preoperative analysis of the volumetric distribution of a face could be performed for the study case, and it could represent a new 3D aesthetic guide in orthognathic surgery.

Developing this concept could bring an interesting paradigm shift in volumetric analysis, corresponding to the most real and specific way to study a face. In the VSP, after marking the occlusion, each operator could choose some favorite reference planes and evaluate the pre- and post-operative volume of the soft tissues segment. New versions of the VSP software could eventually show the skeletal movements and, at the same time, the volumes achieved. With new tools like these, each operator could be able to move the bones functionally to the desired soft tissue volumetric expansion.

We believe that this would be the most practical application of the soft tissues paradigm theory: the predicted facial volumes as a digital tool to plan skeletal movements. Obviously, volumetric cutaneous references would not replace skeletal references, and correct occlusion would always remain a priority; nevertheless, it would be an “extra” tool.

This “Volumetric Cephalometry” could represent, if developed, the evolution of 3D cephalometry.

## 5. Conclusions

In conclusion, we believe that the key to beauty in orthognathic surgery is the achievement of a correct distribution of facial volumes.

With this paper, we propose our aesthetic volumetric references and promote that harmonic balance between facial volumes can be achieved with expansive surgical strategies.

According to our procedural philosophy, the preoperative analysis of volume distribution is a new key point of the planning, and this volumetric analysis could influence the choice of skeletal movements, becoming a new 3D aesthetic guide to surgical planning.

The volumetric cephalometries could become the evolution of 3D cephalometrics, and be introduced into clinical practice as a new tool. We believe that VSP will expand the tools for three-dimensional cephalometry, helping the surgeon to make this new kind of planning in a smarter and faster way.

## Figures and Tables

**Figure 1 jpm-13-00936-f001:**
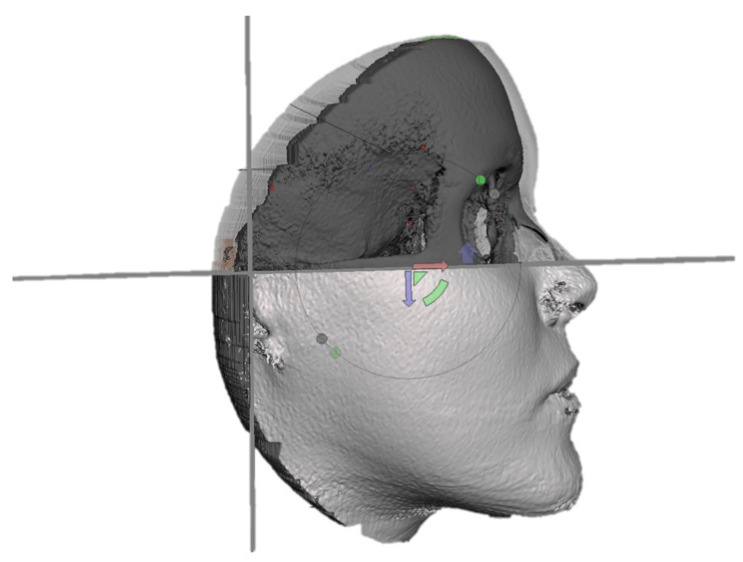
Imaging showing the cutting planes delimiting the lower two-thirds: a superior axial cutting plane passing through FH, and a posterior coronal cutting plane passing through the Porions.

**Figure 2 jpm-13-00936-f002:**
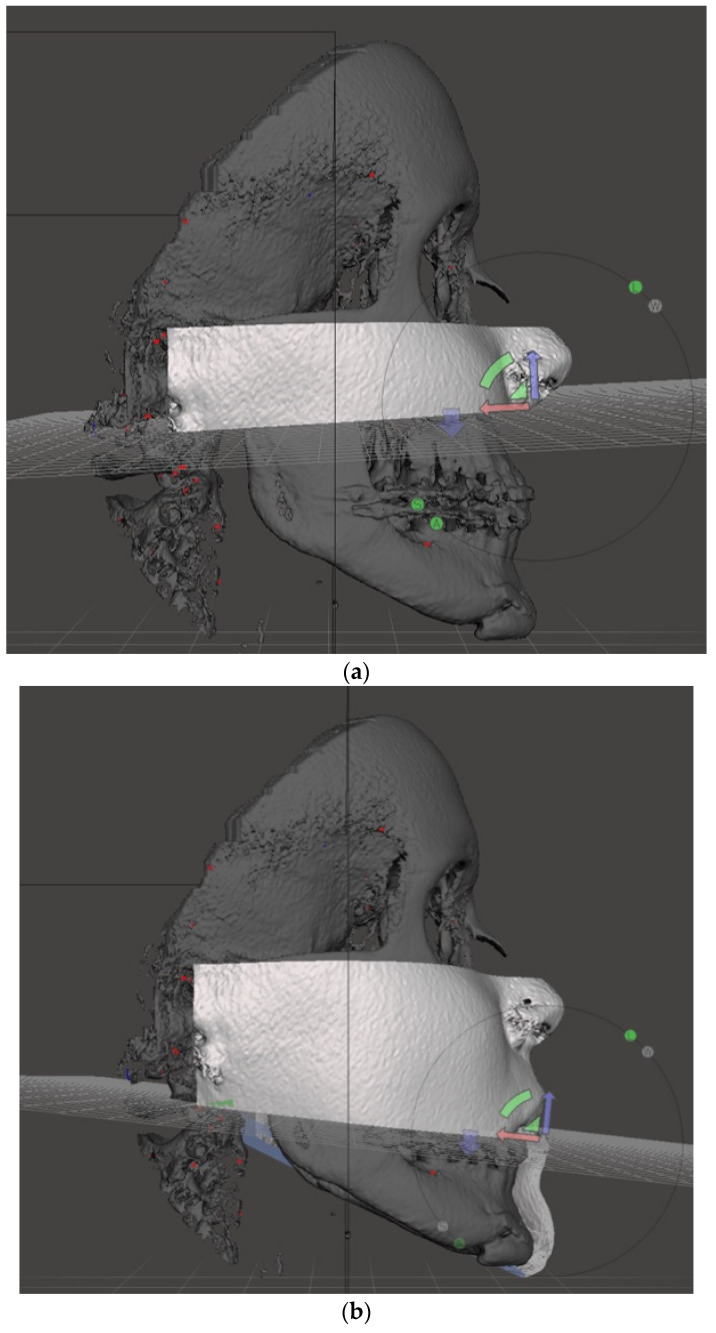
(**a**) Cutting plane passing through the bispinal plane for segmentation of the malar slice. (**b**) Cutting plane passing through the occlusal plane, for segmentation of the maxillary slice. (**c**) Cutting plane passing through the mandibular base for segmentation of the mandibular slice. (**d**) Cutting plane passing through the mental foramina and parallel to FH to separate the chin from the mandibular slice.

**Figure 3 jpm-13-00936-f003:**
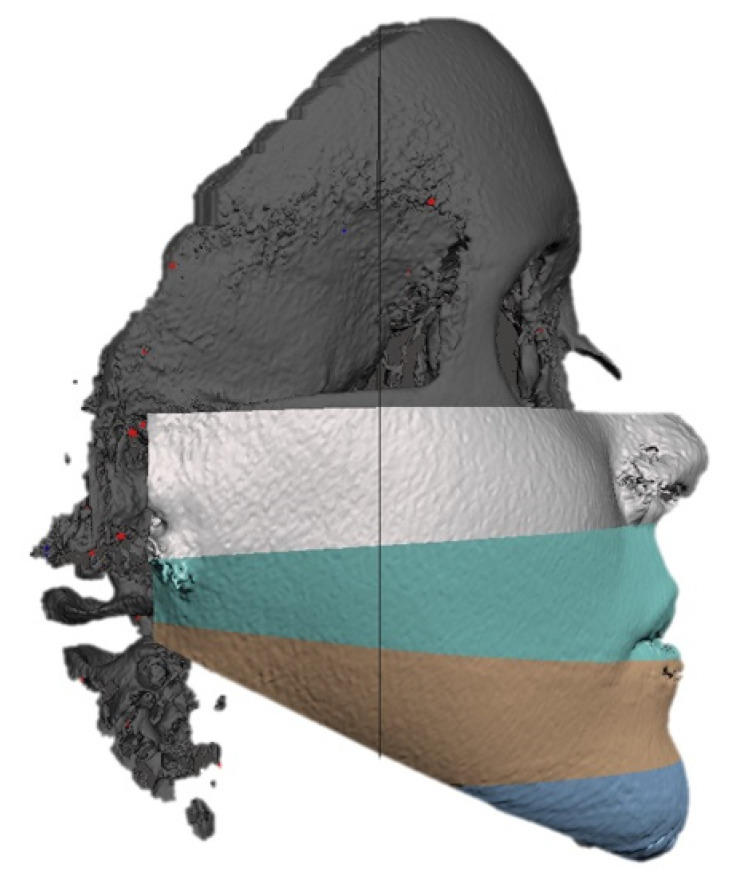
Imaging showing the 4 facial slices after segmentation on CT. From top to bottom: malar, maxillary, mandibular, and chin.

**Figure 4 jpm-13-00936-f004:**
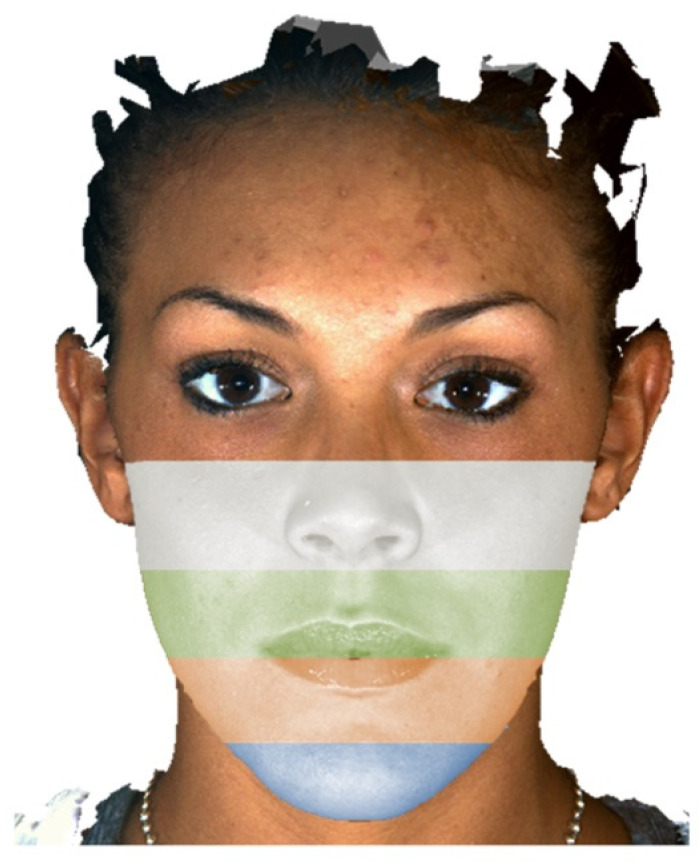
Imaging showing the 4 facial slices after segmentation on 3D photogrammetry. From top to bottom: malar, maxillary, mandibular, and chin.

## Data Availability

Not applicable.

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
