# Peer review of "Is Beauty a Matter of Volume Distribution? Proposal of a New Aesthetic Three-Dimensional Guide in Orthognathic Surgery"

_jpm, 2023, doi:10.3390/jpm13060936_

Round 1

Reviewer 1 Report

The authors discuss an interesting and actual subject. The only thing I reflect on is that the discussion concerning the facial beauty could have been more extensive. An interesting study that I recommend to be published.

Author Response

Dear Editor, 

Please find enclosed the revision of the original manuscript entitled “Is Beauty a matter of volume distribution? Proposal of a new aesthetic three-domensional guide in orthognathic surgery, which we would like to be considered for publication in JPM.

In this work, we analyze the volumetric distribution that we consider ideal in orthognathic surgery, and we propose some of our volumetric references. Furthermore, we discuss the possibility of creating a new 3D Cephalometry, based on facial volumes, where the ideal volumetric redistribution becomes an aesthetic guide in virtual planning.

Nowadays, orthognathic surgery is constantly changing: the aesthetic goal becomes more apparent, rather than just occlusal. More and more importance is given to soft tissues, and not only to the facial skeleton. In addition, preoperative Virtual Surgical Planning (VSP) is increasingly accurate, integrating facial analysis with the 3D-cephalometry on CT. In accordance with this background, we propose a reinterpretation of the beauty canons, based on the study of a ideal volumetric distribution. We have arbitrarily chosen to measure the volumes of these 4 regions, because of the strict relation to the bone segments that are normally moved during orthognathic surgery. Anyhow, a detailed mapping of the mean volumetric distribution in attractive people could be considered: our purpose is basically to show that some volumetric standards could be used as references in the VSP, for each ethnicity. In fact, just like a preoperative 2D or 3D cephalometric analysis of a patient is normally compared to the references to plan bone movements, in the same way, a preoperative analysis of facial volumes could be performed for the study case. By comparing it with the volumetric standards, the surgeon could be guided in planning the most suitable bone movements, once established the occlusion to be achieved.  In other words, this could be a new 3D aesthetic guide in orthognathic surgery.

We believe that this concept, that in our manuscript we call “Volumetric Cephalometry”, is very interesting and innovative. If developed, it could bring to a big paradigm shift in orthognathic planning, since the distribution of volumes corresponds to the most real, specific, and topical way to study a face.

Compared to the first submission, in this new revision, we have increased the Introduction and we have removed some self-referential references, as suggested. Some references, which may seem unrecent, are actually considered historical and milestones in orthognathic surgery.  We have not included clinical cases because this is understood as the first work of a long series, where we discuss in a theoretical way the paradigm shift that we propose, and the cultural background that determines it. We aim to write subsequent works, where we will show the numerous practical applications, showing various clinical cases. For the same reason, the results may seem brief upon initial evaluation. However, as also discussed in the discussion, our goal was not to make just a male/female comparison, since it is obvious that faces are different in the two genders; likewise, it is not our purpose to find some average differences between attractive and not, because it is clear that beauty is individual and not just a mathematical pattern: our main objective is to show our volumetric references and our working method, and to promote the procedural philosophy that we discuss, which we believe is revolutionary.

We hope you will recognize the innovation that this article brings, as it is the synthesis of my entire career in orthognathic surgery, and is the first manuscript of my volumetric philosophy. Recently I have talked a lot about Volumetric Cephalometry, with many international colleagues and Professors, and they all are very interested in knowing, applying and sharing my innovations. We hope you will consider our paper suitable for publication.

All the data presented in this manuscript are original, unpublished, and not concurrently considered by another journal.

All the co-authors have read and approved the final version of the manuscript, and declare that there are no conflicts of interest.   

We look forward to hearing your decision, and we hope that this paper will meet your interests. 

Yours sincerely, 

Reviewer 2 Report

The authors should harmonize the referencing in the manuscript : references 1 to 5 have not the same presentation as the other ones !

L201 : expanded soft issues, to be trended

8 self-citations /37 references are too many !! The literature must be reviewed and changed !!

A clinical case showing how the authors process, should be inserted to easier explain to the reader how it works (the planning of an orthognathic surgery)  !

The idea is not new, but the application to the orthognathic surgery, well ! Proficiat for this !

Author Response

Dear Editor, 

Please find enclosed the revision of the original manuscript entitled “Is Beauty a matter of volume distribution? Proposal of a new aesthetic three-domensional guide in orthognathic surgery, which we would like to be considered for publication in JPM.

In this work, we analyze the volumetric distribution that we consider ideal in orthognathic surgery, and we propose some of our volumetric references. Furthermore, we discuss the possibility of creating a new 3D Cephalometry, based on facial volumes, where the ideal volumetric redistribution becomes an aesthetic guide in virtual planning.

Nowadays, orthognathic surgery is constantly changing: the aesthetic goal becomes more apparent, rather than just occlusal. More and more importance is given to soft tissues, and not only to the facial skeleton. In addition, preoperative Virtual Surgical Planning (VSP) is increasingly accurate, integrating facial analysis with the 3D-cephalometry on CT. In accordance with this background, we propose a reinterpretation of the beauty canons, based on the study of a ideal volumetric distribution. We have arbitrarily chosen to measure the volumes of these 4 regions, because of the strict relation to the bone segments that are normally moved during orthognathic surgery. Anyhow, a detailed mapping of the mean volumetric distribution in attractive people could be considered: our purpose is basically to show that some volumetric standards could be used as references in the VSP, for each ethnicity. In fact, just like a preoperative 2D or 3D cephalometric analysis of a patient is normally compared to the references to plan bone movements, in the same way, a preoperative analysis of facial volumes could be performed for the study case. By comparing it with the volumetric standards, the surgeon could be guided in planning the most suitable bone movements, once established the occlusion to be achieved.  In other words, this could be a new 3D aesthetic guide in orthognathic surgery.

We believe that this concept, that in our manuscript we call “Volumetric Cephalometry”, is very interesting and innovative. If developed, it could bring to a big paradigm shift in orthognathic planning, since the distribution of volumes corresponds to the most real, specific, and topical way to study a face.

Compared to the first submission, in this new revision, we have increased the Introduction and we have removed some self-referential references, as suggested. Some references, which may seem unrecent, are actually considered historical and milestones in orthognathic surgery.  We have not included clinical cases because this is understood as the first work of a long series, where we discuss in a theoretical way the paradigm shift that we propose, and the cultural background that determines it. We aim to write subsequent works, where we will show the numerous practical applications, showing various clinical cases. For the same reason, the results may seem brief upon initial evaluation. However, as also discussed in the discussion, our goal was not to make just a male/female comparison, since it is obvious that faces are different in the two genders; likewise, it is not our purpose to find some average differences between attractive and not, because it is clear that beauty is individual and not just a mathematical pattern: our main objective is to show our volumetric references and our working method, and to promote the procedural philosophy that we discuss, which we believe is revolutionary.

We hope you will recognize the innovation that this article brings, as it is the synthesis of my entire career in orthognathic surgery, and is the first manuscript of my volumetric philosophy. Recently I have talked a lot about Volumetric Cephalometry, with many international colleagues and Professors, and they all are very interested in knowing, applying and sharing my innovations. We hope you will consider our paper suitable for publication.

All the data presented in this manuscript are original, unpublished, and not concurrently considered by another journal.

All the co-authors have read and approved the final version of the manuscript, and declare that there are no conflicts of interest.   

We look forward to hearing your decision, and we hope that this paper will meet your interests. 

Yours sincerely, 

Professor Alberto Bianchi